# Perceived listening ability and hearing loss: Systematic review and qualitative meta-synthesis

**Sarah E. Hughes**[1,2,3,4,5,6]*, **Isabelle Boisvert**[7,8], **Catherine M. McMahon**[8,9], **Anne Steyn**[10], **Katie Neal**[8,9,11,12]

**1** Centre for Patient Reported Outcome Research, Institute of Applied Health Research, University of Birmingham, Birmingham, United Kingdom, **2** National Institute for Health and Care Research (NIHR) Applied Research Collaboration (ARC), West Midlands, United Kingdom, **3** Birmingham Health Partners Centre for Regulatory Science and Innovation, University of Birmingham, Birmingham, United Kingdom, **4** National Institute of Health and Care Research (NIHR) Blood and Transplant Research Unit (BTRU) in Precision Therapeutics, University of Birmingham, Birmingham, United Kingdom, **5** Narra Consulting Limited, Wales, United Kingdom, **6** Faculty of Medicine, Health, and Life Science, Swansea University, Swansea, United Kingdom, **7** Faculty of Medicine and Health, University of Sydney, Sydney, NSW, Australia, **8** HEAR Centre, Macquarie University, Sydney, Australia, **9** Department of Linguistics, Macquarie University, Sydney, NSW, Australia, **10** Citizen Partner, Sydney, NSW, Australia, **11** The Shepherd Centre, Sydney, NSW, Australia, **12** The Listening Lab, Sydney, NSW, Australia

* s.e.hughes@bham.ac.uk

**Data Availability Statement:** All relevant data are within the paper and its Supporting Information files.

## Abstract

### Background

Hearing loss (HL) can affect communication in complex ways. Understanding how adults with HL reflect on and conceptualise the way they listen (metacognition) is required if interventions, and the outcome measures used to evaluate them, are to address barriers to functional communication arising from HL.

### Objectives

This study describes how adults with HL experience and report the processes, behaviours, and components of listening, as presented in published studies.

### Design

Systematic review and meta-synthesis of qualitative studies.

### Methods

Systematic searches identified English-language, peer-reviewed journal articles reporting the results of qualitative or mixed-methods studies of adults' with HL perceived listening abilities. Medline, PsychInfo, Web of Science, Embase, and Google Scholar were searched from inception to November 2021. Handsearching reference lists of included studies identified additional studies for inclusion. The Critical Appraisal Skills Programme (CASP) qualitative checklist was used to appraise studies' methodological quality. Data from included studies were analysed using thematic meta-synthesis. The Grading of Recommendations

**Funding:** This work was funded by Cochlear Limited (Funder Ref: Functional Listening for Communication Project). The funders had no role in study design, data collection and analysis, decision to publish, or preparation of the manuscript.

**Competing interests:** We declare that Dr Hughes receives funding from the National Institute for Health Research (NIHR) Applied Research Collaboration (ARC) West Midlands and UK Research and Innovation (UKRI), UK SPINE, and declares personal fees from Aparito Limited, CIS Oncology, and Astra Zeneca outside the submitted work. Dr Boisvert declares funding from Birmingham Health Partners Centre for Regulatory Science & Innovation / UK SPINE outside the submitted work. All other authors have no competing interests to declare. This does not alter our adherence to PLOS ONE policies on sharing data and materials.

Assessment, Development and Evaluation (GRADE) Confidence in the Evidence from Reviews of QUALitative (CERQual) approach assessed confidence in the review findings. Two reviewers independently completed all screening and quality appraisal. Thematic meta-synthesis and GRADE CERQual assessment was completed by one reviewer and confirmed by a second reviewer. Discrepancies were resolved through discussion.

## Results

Data from 46 studies were included in the review. Thematic meta-synthesis identified six descriptive themes: 1) perceived listening ability; 2) external modifiers; 3) psychosocial impacts of hearing loss; 4) communication partner perspectives; 5) self-efficacy for listening; and 6) cognitive load. GRADE CERQual ratings for descriptive themes ranged from low to moderate confidence. Descriptive themes were related by analytic themes of liminality and reciprocity.

## Conclusions

Adults with HL provide in-depth accounts of components and processes of listening, with studies reporting both cognitive and affective experiences consistent with theoretical models of metacognition. The findings will inform content generation for a hearing-specific patient-reported outcome measure of perceived listening ability in everyday communication.

## Introduction

Communication is a shared activity representing the coordinated actions of two or more people to achieve a common goal [1]. Successful communication involves a dynamic complex of hearing, listening, language and cognition that can be significantly disrupted when a hearing loss (HL) is present. Despite widespread recognition of the impact of HL on listening and communication, clinical assessment in the hearing clinic has focussed on the measurement of the HL in isolation (e.g., pure tone thresholds and speech reception measures) and outside of the communicative context in which listening typically occurs [2].The reductionist nature of this approach has consequences for the evaluation of listening in the context of HL and how interventions to mitigate the impacts of HL are designed and delivered.

An alternative, ecologically valid, and patient-centred approach to outcome measurement in HL is to consider the measurement of listening ability from the perspective of the everyday communication experience of individuals with HL. In this context, self-report has potential for assessing how an individual feels and functions in the listening situations of everyday communication. Adopting a more person-centred approach that is cognizant of the interactive, multimodal and highly contextualised nature of listening may offer insights that inform novel interventions for HL, and novel ways to assess the benefits of these interventions.

Patient-reported outcomes (PROs), as measures of health and functioning reported directly by the patient, hold promise as a means of achieving a more holistic measurement of listening for communication. PROs are recommended for use as endpoints in clinical trials, for real-world evidence generation, and have shown benefits when used routinely in clinical practice [3–8]. Regulatory guidance exists to support the development and validation of PROs and includes guidance on concept elicitation [3,4]. Foremost, items must reflect the construct of interest and be relevant and understood as intended by the target population. For items to be relevant and comprehensible, developers must be able to discern which latent concepts relating

to the target construct are perceivable (i.e., within conscious awareness) by members of the target population and suitable for operationalisation (i.e., whether individuals are able to recognise and appraise the concept, retrieve the required information from memory, judge the applicability of the retrieved information, and estimate a response on a pre-specified scale) [9–11].

Theoretical models, when used in conjunction with first-hand accounts of the target construct, can support the framing of a PRO's assessment [11]. In the context of perceived listening ability for functional communication, theories of metacognition could play a supportive role in concept elicitation and generation of a conceptual framework and measurement model. Metacognition refers to "the human ability to be conscious of one's mental process" [12–14]. Models of meta-cognition, notably Flavell's seminal Model of Cognitive Monitoring, have been extensively applied within education, particularly in relation to second language listening [15]. Flavell's model describes meta-cognition in terms of the actions and interactions of four distinct phenomena: 1) metacognitive knowledge (defined as the segment of an individual's stored world knowledge that "consists primarily of knowledge of beliefs about what factors or variables act and interact in what ways to affect the course and outcome of cognitive enterprises" (p. 907); 2) metacognitive experiences (any conscious cognitive or affective experiences related to a cognitive task); 3) goals/tasks, and 4) actions/strategies [12]. Metacognitive knowledge may be further classified as knowledge relating to persons, tasks, or strategies. In the context of listening (as a cognitive activity), person knowledge could includer an individual's judgements about their listening abilities and stored knowledge about internal and external factors likely to affect listening success. Task knowledge could include knowledge relating to the purpose, demands, and nature of a listening task. Task knowledge enables individuals to appraise the difficulty of a listening activity to support strategy selection and deployment [12,16,17]. Strategy knowledge may be defined as knowledge useful for achieving listening and/or communication goals.

Conceptualising perceived listening ability as a special type of meta-cognition could support the operationalisation of the concept for measurement as a PRO. A meta-cognitive lens could help to identify domains of perceived listening ability within conscious awareness for adults with HL thereby rendering these concepts candidates for measurement. For example, in the field of second language (L2) learning, Vandergrift et al. (2008) used Flavell's model to underpin the development of the Metacognitive Awareness Listening Questionnaire (MALQ), a self-report instrument designed to assess the extent to which language learners are aware of and can regulate the process of L2 listening comprehension [16]. In relation to HL, metacognition has been studied primarily in children within educational settings. Studies have explored metacognitive awareness during classroom instruction and the impact of metacognitive strategy deployment on reading behaviour and comprehension in deaf and hard-of-hearing students [18,19]. With the exception of studies exploring the role of metacognitive beliefs on distress in tinnitus, comparatively few studies have studied the metacognition of listening explicitly in adults with HL [20]. However, a rich qualitative literature is available relating variously to metacognitive concepts such as the influence of person, task and strategy factors on listening performance in the context of the lived experience of HL. This systematic review and qualitative meta-synthesis use this literature to describe first-hand accounts of adults' with HL listening-related knowledge and experience.

## Methods

We reported this systematic review and meta-synthesis in line with the PRISMA guidelines and the Joanna Briggs Institute guidance for systematic reviews of qualitative evidence [21].

The protocol was registered on PROSPERO (Registration number: CRD42020213389), the international prospective register of systematic reviews, and published in a peer-reviewed journal [22].

## Search strategy

The search strategy was developed in Medline (Ovid) using terms based on the population (i.e., adults with HL), study design, and terms related to listening (including MeSH terms and their synonyms). The full Medline search strategy is in S1 File. We searched Medline (Ovid), PsychInfo (Ovid), Embase (Ovid), Web of Science, and Google Scholar (first 200 records) from inception to 15th November 2021. Database selection, including Google Scholar, was guided by recommendations for optimal yields proposed by Bramer et al [23]. We hand-searched the references lists of the included studies.

## Eligibility criteria

Eligible studies were English-language, peer-reviewed manuscripts of primary qualitative or mixed-methods studies that reported first-hand accounts from adults ($\geq$18 years of age) with HL and/or their communication partners on their experiences of listening for communication in daily life. Quantitative studies, studies involving or reporting on children as participants, editorial, opinions, and letters were excluded. Only the qualitative data from mixed-methods studies were analysed. No restrictions were placed on the date of publication, country of origin, setting, degree and aetiology of HL, or hearing device. We excluded studies reporting solely on the psychosocial aspects of HL but included those studies that discussed psychosocial impacts of HL alongside an exploration of perceived listening ability. Studies reporting on sign language were excluded. The rationale for the study eligibility criteria is described in detail in the study protocol [22].

## Study selection

We uploaded the search results to Endnote (version 9.3, Clarivate Analytics). After duplicates were removed, the remaining records were imported to Covidence web-based systematic review software (www.covidence.org) for screening and data extraction. Two reviewers (KN, SH) independently screened all titles and abstracts and full text articles for eligibility. Reasons for exclusion were documented at the full text screening stage. Disagreements were resolved through discussion or by a third author (IB).

## Assessment of the methodological quality of the included studies

Two reviewers (SH, KN) used the Critical Appraisal Skills Programme (CASP) checklist for qualitative research to independently assess the methodological quality of the included studies. Differences were resolved through discussion or by involving a third reviewer. Percentage agreement, Cohen's kappa ($\kappa$), as a measure of agreement adjusted for chance, and an intra-class correlation coefficient (ICC) was calculated to establish inter-rater agreement and reliability respectively [24,25]. A kappa value of 0.69–0.79 was considered evidence of "moderate" agreement and $\kappa \geq 0.80$ represented "good" agreement. ICC > 0.75 was indicative of good inter-rater reliability [26].

## Data extraction

We used Covidence to extract the following information from the included articles: author, year of publication, country of origin, study aims and objectives, population characteristics,

setting, phenomena of interest, methodological approach, and data collection and analysis methods.

## Data synthesis

We uploaded the full-text articles to NVivo (Version 1.5) qualitative data analysis software and analysed the results sections (including both narratives and participant quotes) and relevant supplementary material from the included studies. We used the methodology proposed by Thomas and Harden to conduct the thematic meta-synthesis [27]. First, we completed line-by-line coding to label concepts presented in the findings which we then compared across the included studies, grouping these initial codes into higher-level descriptive themes. To provide a rich, nuanced description of the descriptive themes, we used axial coding to develop a further coding paradigm to describe a set of characteristics (subcategories) operating within each descriptive theme [28]. The descriptive themes were then interrogated to reflexively construct a hypothesised model of the proposed relationships amongst the themes. One researcher (SH) completed the inductive line-by-line initial coding. Where relevant, multiple codes were assigned to a single data unit. The coding for ten percent of included studies (n = 5) was reviewed for accuracy by a second researcher (KN). The studies were selected randomly for review using a web-based random number generator. Themes were co-constructed through discussion between the authors (KN, SEH). The discussions were supported by coding and analytical memos that documented coding decisions and theme development. Both authors used reflexive memos to document the researcher lenses which informed their engagement with the analytic process. SH is a PRO researcher and qualified speech and language therapist. KN is a qualified audiologist, hearing researcher, and practicing clinician with lived experience of HL.

## Assessment of confidence in the synthesis findings

The Grading of Recommendations Assessment, Development and Evaluation–Confidence in Evidence from Reviews of Qualitative research (GRADE-CERQual) approach was used to establish the degree of confidence that can be applied to the individual, descriptive findings [29]. The data supporting each of the six descriptive themes were appraised against the GRADE-CERQual criteria for: 1) methodological limitations; 2) relevance; 3) coherence; and 4) adequacy. An overall rating of confidence of "high", "moderate", "low" or "very low" was assigned to each finding (theme), considering each of the four GRADE CERQual components described in Table 1 [30–34].

**Table 1. Overview of the four components of the GRADE-CERQual approach for assessing confidence in findings of reviews of qualitative research.**

| GRADE-CERQual Component | Definition |
| --- | --- |
| Methodological limitations | "The extent to which there are concerns about the design or conduct of the primary studies supporting a review finding." [32] (p.25) |
| Coherence | "An assessment of how clear and cogent that fit is between the data from the primary studies and a review finding that synthesises the data." [30] (p.35) |
| Adequacy | "The degree of richness and quantity of data supporting a review finding." [33] (p.45) |
| Relevance | "The extent to which the body of data from the primary studies supporting a review finding is applicable to the context specified in the review question." [31] (p.53) |

The aggregated CASP checklist results for the contributing studies were used to assess the methodological limitations of the contributing studies per synthesis finding. Synthesis findings were assessed as having "no or very minor concerns" for coherence if there was "good and consistent fit" between the theme and data from the contributing studies [30,35]. Synthesis findings were assessed as having "no or very minor concerns" for adequacy if more than 50% of the contributing studies contributed to a theme and the data from these studies were sufficiently rich [36]. Synthesis findings were assessed as having "no or very minor concerns" for relevancy if most of the papers included in a descriptive theme address the review question overtly (i.e., perceived listening ability), most participants were adults with HL and their communication partners, and findings addressed listening in everyday communication settings. For all GRADE CERQual components, appraisal began with the assumption that there were no concerns regarding the body of data contributing to each finding with the rating downgraded based on appraisal results. The GRADE CERQual approach was not applied to analytic themes due to the interpretative nature of these higher-order themes [37].

## Results

The search outcomes are reported in detail in the Preferred Reporting Items for Systematic Reviews and Meta-analyses (PRISMA) flow chart (Fig 1). In total, 3,214 records were identified for title and abstract screening. Of these, 2,803 were excluded and 137 underwent full-text review. Forty-six studies met the study eligibility criteria and were included in the review.

### Study characteristics

Included studies were published in seven high-income countries (Australia, Canada, Ireland, Norway, Sweden, United Kingdom, USA) and one middle-income country (Brazil) between 1967 and 2021, with 30 studies published since 2010. Mean sample size was 26.4 participants (range = 3–207, SD = 33.4). Five studies (10.9%) included communication partners (e.g., spouses, adult children) and/or healthcare practitioners in the study sample. Twenty-four (52.2%) studies explored the lived experience of HL, ten (21.7%) studies examined adults' with HL experiences with audiological rehabilitation and healthcare consultations, two studies (4.3%) explored listening effort and listening-related fatigue, four studies (8.7%) examined hearing device use, three studies (6.5%) explored adults' with HL musical experiences, and three studies (6.5%) reported findings to inform the development of patient-reported outcome measures (PROMs). None of the studies specifically explored perceived listening ability or adults' with HL meta-cognitive awareness of listening. Most studies used interviews (n = 27, 58.7%) as a data collection method. Seven studies (15.2%) used focus groups, three studies (6.5%) used survey techniques and nine studies (19.6%) used a multi-methods approach. Data analysis methods used most often were thematic analysis (n = 15, 32.6%), content analysis (n = 6, 13.0%), and grounded theory (n = 13, 28.3%). Characteristics of the included studies are summarised in S1 Table.

### Methodological quality of studies

None of the included studies were excluded following appraisal of their methodological quality. Table 2 presents a summary of the quality appraisal using the CASP assessment tool [38]. Forty-three (93.5%) studies had a clear statement of the aims of the research, and 45 (97.5%) used an appropriate qualitative methodology. Most used an appropriate research design (n = 41, 89.1%), recruitment strategy (n = 38, 82.6%), data collection (n = 44, 95.7%), analysis methods (n = 32, 69.6%), and provided a clear statement of the findings (n = 42, 91.3%). A lack of reporting of ethical issues (n = 19, 41.3%), with few studies (n = 13; 28.3%) explicitly

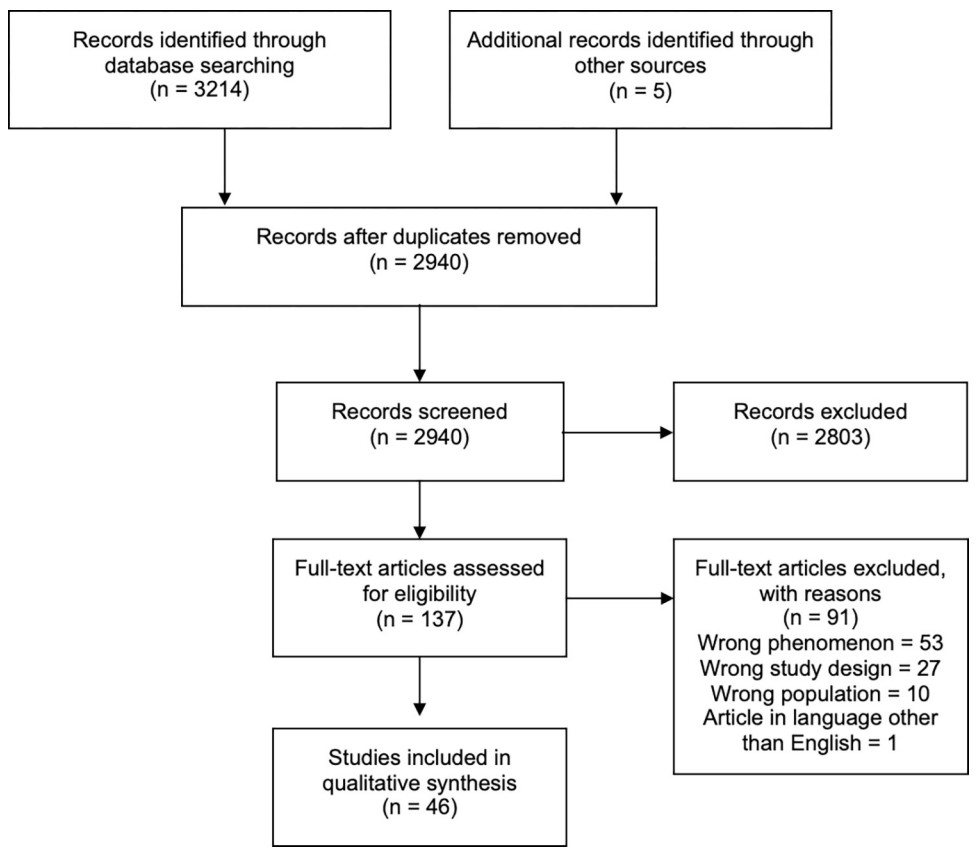

**Fig 1. PRISMA flow diagram.**

addressing the researcher-participant relationship, were commonly identified weaknesses of the included studies.

Two reviewers (SEH, KN) appraised independently the methodological quality of all studies. Cohen's kappa was 0.855 (p < 0.001), suggesting "strong" inter-rater agreement. ICC was 0.92 (95% CI 0.91–0.94, p < 0.001), indicating excellent inter-rater reliability.

### Thematic synthesis

The descriptive themes of this meta-synthesis highlight the overt ideas in the data and therefore are descriptive rather than interpretative in their nature. The six descriptive themes relate broadly to: 1) listening abilities (or behaviours) which are perceivable by adults with HL when listening in the communication situations of daily life; 2) external elements which influence listening capability, ability and behaviour; 3) the wider psychosocial impacts of HL attributed by participants to listening; 4) perspectives, roles, and implications of communication partners on the listening abilities of adults with HL; 5) the role of self-efficacy; and 6) cognitive load. The themes include data extracted to 212 codes from all the included studies (n = 46). A brief description of each theme is presented below. Table 3 presents exemplar extracts and coding of the original studies for each descriptive theme.

**Descriptive theme 1: Perceived listening ability as navigation (within the dynamic, transactional landscape of communication).** Data from 43 (93.5%) studies contributed to this theme which describes self-reported listening ability from the perspective of adults with HL. These data suggested listening is perceived by adults with HL as a *navigational act*, a series

**Table 2. Results of CASP quality appraisal (N = 46).**

| Study ID | 1. Was there a clear statement of the aims of the research? | 2. Is a qualitative methodology appropriate? | 3. Was the research design appropriate to address the aims of the research? | 4. Was the recruitment strategy appropriate to the aims of the research? | 5. Was the data collected in a way that addressed the research issue? | 6. Has the relationship between the researcher and participants been adequately considered? | 7. Have ethical issues been taken into consideration? | 8. Was the data analysis sufficiently rigorous? | 9. Is there a clear statement of findings? |
|---|---|---|---|---|---|---|---|---|---|
| Alsawy 2020 | 1 | 1 | 1 | 1 | 1 | 0 | ? | 1 | 1 |
| Athalye 2014 | 1 | 1 | 1 | 1 | 1 | 0 | 1 | 0 | 1 |
| Bennion 2013 | 1 | 1 | 1 | 1 | 1 | ? | 0 | ? | 1 |
| Bennett 2021 | 1 | 1 | 1 | 1 | 1 | 1 | 1 | 1 | 1 |
| Bryant 2020 | 1 | 1 | 1 | 1 | 1 | 1 | 1 | 1 | 1 |
| Choi 2018 | 1 | 1 | 1 | 1 | ? | 0 | 0 | ? | 1 |
| Cudmore 2017 | 0 | 0 | 0 | 1 | 0 | 0 | ? | ? | 0 |
| Davies 2001 | 1 | 1 | 1 | 1 | 1 | ? | ? | 0 | 1 |
| Davis 2021 | 1 | 1 | 1 | 1 | 1 | ? | 1 | 1 | 1 |
| Dawes 2014 | 1 | 1 | 1 | 1 | 1 | ? | ? | 1 | 1 |
| Fitzpatrick 2010 | 1 | 1 | 0 | 1 | 1 | ? | ? | 1 | 1 |
| Foster 2003 | 1 | 1 | 1 | 0 | 1 | ? | ? | 1 | 1 |
| Fulford 2011 | 1 | 1 | 1 | 1 | 1 | ? | ? | 1 | 1 |
| Funk 2018 | 1 | 1 | 1 | 0 | 1 | ? | 0 | 0 | 1 |
| Gfeller 2019 | 1 | 1 | 1 | 1 | 1 | ? | 1 | 1 | 1 |
| Giolas 1967 | 1 | 1 | 1 | ? | 1 | ? | ? | 1 | 1 |
| Granberg 2014 | 1 | 1 | 1 | ? | 1 | ? | 1 | 1 | 1 |
| Hallam 2008 | 1 | 1 | 1 | 1 | 1 | 1 | ? | 1 | 1 |
| Hallberg 1991 | 1 | 1 | 1 | 0 | 1 | ? | ? | 1 | 1 |
| Hallberg 1993 | 1 | 1 | 1 | 1 | 1 | ? | ? | ? | 1 |
| Hallberg 1995 | 1[a] | 1 | 1 | 1 | 1 | ? | ? | 1 | 1 |
| Hallberg 1996 | 1 | 1 | 0 | 1 | 1 | ? | ? | 0 | 1 |
| Hass-Slavin 2005 | 1 | 1 | 1 | 1 | 1 | ? | ? | ? | 1 |
| Heacock 2019 | 1 | 1 | 1 | 1 | 1 | ? | ? | 1 | 1 |
| Hua 2015 | 1 | 1 | 1 | 1 | 1 | 1 | 1 | 1 | 1 |
| Hughes 2018 | 1 | 1 | 1 | 1 | 1 | 1 | 1 | 1 | 1 |
| Iezzoni 2004 | 1 | 1 | 1 | ? | 1 | ? | ? | ? | 1 |

*(Continued)*

**Table 2.** (Continued)

| Study ID | 1. Was there a clear statement of the aims of the research? | 2. Is a qualitative methodology appropriate? | 3. Was the research design appropriate to address the aims of the research? | 4. Was the recruitment strategy appropriate to the aims of the research? | 5. Was the data collected in a way that addressed the research issue? | 6. Has the relationship between the researcher and participants been adequately considered? | 7. Have ethical issues been taken into consideration? | 8. Was the data analysis sufficiently rigorous? | 9. Is there a clear statement of findings? |
|---|---|---|---|---|---|---|---|---|---|
| Jeffs 2015 | 1 | 1 | 1 | 1 | 1 | 1 | 1 | 1 | 1 |
| Jonsson 2018 | 1 | 1 | 1 | 1 | 1 | 1 | 1 | 1 | 1 |
| Laplante-Levesque 2006 | 1 | 1 | ?[c] | 1 | 1 | ? | 1 | 1 | 1 |
| Lockey 2010 | 1 | 1 | 1 | 1 | 1 | 1 | 1 | 1 | 1 |
| Lucas 2018 | 1 | 1 | 1 | 1 | 1 | ? | ? | 1 | 1 |
| McRackan 2017 | 0[b] | 1 | 1 | 1 | 1 | 0 | 0 | 0 | 0 |
| Miller 2017 | 1 | 1 | 1 | 1 | 1 | 1 | ? | 0 | 0 |
| Preminger 2014 | 1 | 1 | 1 | 1 | 1 | 1 | ? | 1 | 1 |
| Pryce 2012 | 1 | 1 | 1 | 1 | 1 | ? | 1 | ? | 0 |
| Punch 2019 | 1 | 1 | 1 | 1 | 1 | ? | 1 | 1 | 1 |
| Rembar 2009 | 1 | 1 | 1 | 1 | 1 | ? | 1 | ? | 1 |
| Scarinci 2008 | 1 | 1 | 1 | 1 | 1 | ? | 1 | 1 | 1 |
| Scarinci 2009 | 1 | 1 | 1 | 1 | 1 | ? | 1 | 1 | 1 |
| Scharp 2020 | 1 | 1 | 1 | 1 | 1 | 1 | 1 | 1 | 1 |
| Schlau 2004 | 0 | 1 | ? | 1 | 1 | 1 | ? | 1 | 1 |
| Shaw 2013 | 1 | 1 | 1 | ? | 1 | ? | ? | 1 | 1 |
| Tye-Murray 2009 | 1 | 1 | 1 | 1 | 1 | ? | ? | 1 | 1 |
| Vaisberg 2019 | 1 | 1 | 1 | 1 | 1 | 1 | 1 | 1 | 1 |
| Vieira 2018 | 1 | 1 | 1 | ? | 1 | ? | ? | 1 | 1 |
| Total Yes n (%) | 43 (93.5%) | 45 (97.8%) | 41 (89.1%) | 38 (82.6%) | 44 (95.7%) | 13 (28.3%) | 19 (41.3%) | 32 (69.6%) | 42 (91.3%) |

[a]"1" = Yes.

[b] "0" = No.

[c]"?" = Unclear.

of negotiations, both with oneself and with others. Describing listening as navigation emphasises the concepts of motion and energy, an acknowledgment of the transactional and fluid nature of communication. The actions of an individual reflect a dynamic, choice-making exercise that individuals engage as they steer a course through a communicative interaction. Listening acts were described in different ways by the body of studies. Data from 29 studies suggested that listening ability or behaviour could be classified as *adaptive* or *maladaptive*, meaning it either facilitated or impeded interaction [22,39,41–52,54,56–70]. In 12 studies

**Table 3. Development of the descriptive and analytic themes showing examples of the initial codes comprising the descriptive themes with supporting exemplar quotes.**

| Descriptive Theme | Subtheme | Initial codes | Exemplar quotes |
|---|---|---|---|
| D1. Perceived listening ability as navigation (within the dynamic, transactional space of communicative acts) | Skill focus as listening-specific or compensatory | Use of written text | *". . .if it's important they are going to e-mail me or give me the minutes."—participant* [39] |
| | | Ability to filter sound | *". . .The brain can actually do so when you are not hearing impaired, it can sort out the things that do not concern what you are talking about. Even if 20 people are sitting around you, the brain can do that. It is the brain that does that, but it can only do so if you hear well." (DK, M, 62, Profile A)"* [40] |
| | Orientation to self or other | Bringing in a third person | *"I always have to go to with husband. My husband needs to translate [speak loudly and repeatedly] for me." -participant 012* [41] |
| | Adaptive/maladaptive | Bluffing | *"But he'll say something, and I'll ask him again. And he'll say it about the same tone, so I'll just laugh and nod, and [I] never heard it.. . . Oh yeah, missed the whole thing"–participant* [42] |
| | | Controlling the conversation | *". . .I have to be in control of things just so I know what is going on. (14D)"—participant* [43] |
| | | Controlling the conversation | *"This strategy included initiating or dominating conversations in order to control conversation theme."—authors* [42] |
| D2. External modifiers of perceived listening ability | Complexity | Complexity | *"The greater or lesser understanding will depend on numerous factors, such as positioning the phone correctly in the processor microphone; presence of background noise; volume, speed and timbre of voice; sound quality of the telephone; whether it is a familiar voice or one they are not accustomed to; training on the functional use of this skill."—authors* [44] |
| | Environmental modifiers | Distance | *"I couldn't understand or hear from a distance the interaction between customers. . ."* [45] |
| | | Acoustics of the environment | *"They put new insulation on the ceiling that is very absorbent. They also put down a large mat that absorbs quite a lot of noise. I feel that there's a huge difference between working in my classroom and going into somebody else's classroom that has not been soundproofed. I think the pupils think so too. There is a big difference between being in a facilitating environment and an aggravating one." (No. 4).* [46] |
| | Speaker characteristics | Multiple speakers | *"I love going out for dinner where there's max four people at a table, any more than that and you think you can wipe half the conversation off because you can't hear them."—participant* [47] |
| | | Ability to listen to soft speech | *"Informants mentioned that they worried when they did not hear soft-spoken words said in confidence by their children. . ."—authors* [48] |
| | Use of technical supports | Use of subtitles | *". . .'the television is kept permanently on subtitles and that makes life possible really, I mean even listening with HAs on, listening to a news bulletin I tend to miss the detail and subtitles are very useful. . ."—participant* [49] |
| | | Listening tactics–technical supports | *"Several participants reported using technology to communicate with family and friends (i.e., using text messaging instead of talking in person and using voice recorders to record sound and playing it louder)."—authors* [41] |
| | Information complexity | Ability to understand short v long units of speech | *". . . for me it's, when people give, you know, longer sentences, talking longer, then I can eh, get more information [get the gist of the communication partner's message], you know."—participant* [50] |

*(Continued)*

**Table 3.** (Continued)

| Descriptive Theme | Subtheme | Initial codes | Exemplar quotes |
|---|---|---|---|
| D3. Psychosocial impacts of HL | Social activity | Participation | "I find it very frustrating when my hearing goes down and I can't communicate with students, I cannot participate properly in management decisions." (P1, a university lecturer) [51] |
| | | Withdraw from conversation | "One participant's solution to this problem was to no longer attend the theatre and to cut herself off from the activity 'you think right, well I won't bother going, It's a waste of time, so that's a shame cos you start to cut yourself off from certain things because of it [HI]' (Female 1)." [49] |
| | Emotional and physical wellbeing | Impact of HL on emotional health | She told me. . . "I am tired of you writing notes. Get out of here. Go away". . . I packed up my stuff and cried." [45] |
| | Quality of life | Quality of life | "For CI users, being able to hear makes them feel capable and valid again. It generates self-confidence, increases self-esteem, and arouses motivation, the will to do things and live. All this improvement in their quality of life (QoL) minimizes the impact of disability, leading them to look at deafness and life from a more positive perspective."[44] |
| D4. Communication partner perspectives, roles, and implications | Roles and adaptations | Advocating for partner with HL | "The adult children viewed their role in HHC primarily as communication management." [52] |
| | | Using communication techniques | "The use of communication techniques to aid their partner's communication was another problem area for spouses, with participants reporting difficulty managing the need to use face-to-face communication, positioning strategies and alternative forms of communication, as well as having to raise the volume of their voice, interpret conversations for their partners and prompt their partners during group conversation. Spouses reported the continual use of these strategies to be draining and therefore they experienced cognitive difficulties managing these tasks." [53] |
| | Perspectives (beliefs and attitudes) | Supporter | "F [communication partner] may feel they are giving support and it is not being appreciated: 'Yes, she does not look for support but she certainly needs support and I give her support. Now, she might argue with that and say I don't support her, I don't know what she would say about that' (8F) . . .." [43] |
| | Wider impacts of HL on relationships | Feeling stressed | "My husband sometimes gets annoyed because I can't hear and he has to keep repeating stuff. . . or he has to come directly to me and he feels that I should be able to hear him.. . . I hate to use the word stress, but— it's just you know, the conflicts sometimes that it creates.. . . I don't want him to be mad at me all the time, you know." [42] (p. 331) |
| D5. Self-efficacy for listening | | Self-efficacy | "When I was looking at other people, I didn't see it [using HAs] as bad but when I saw it on myself it was like, "I'm weak and I need these." And you know, people are going to make fun of me and I've made it a lot bigger of deal than it should have been." [54] |
| | | Being constrained | ". . .subjects have a perception of imposed limitations on the environment, restricted access to information and a feeling of being constrained. . ." [55] |
| | | Perceptions of disability | "Although participants recognised that their HL had reduced their ability to cope with many everyday situations, they did not appear to consider themselves to be in poor health because of their HL." [56] |
| | | Perceptions of disability | ". . .I don't consider myself hearing impaired, even because there are worse disabilities. [. . .] I think of myself as a normal person because I'm trying to improve more and more. . ." [44] |

(*Continued*)

**Table 3.** (Continued)

| Descriptive Theme | Subtheme | Initial codes | Exemplar quotes |
|---|---|---|---|
| D6. Cognitive load | | Time lag | *"Just tagging along, harder to contribute because of "listening and assimilating" time, the moment passes and someone else is speaking"—participant 001* [57] |
| | | Attention | *"Being more attentive and concentrating in conversation with others, are other codes grounded in the data: 'II is something you do unconsciously . . . focus your concentration. . . depending on whether it is interesting or not. II is perhaps a way to influence the ability to hear; to listen actively or actively participate in communication or just listen. In that case l hear nothing'."* [58] |

*liminality* (i.e., the threshold of conscious awareness) was suggested to constrain whether a listening-related skill was perceivable by adults with HL (and therefore accessible for insight) and that liminal thresholds varied across skills and across individuals [43,45,46,49,52,58,59,61,70–73]. The contributing studies suggested that listening skills may be further classified as having an *auditory-linguistic focus* (28 studies) [39,40,43,45–47,49–51,56,57,59,60,62–65,69–72,74–79], or a *compensatory focus* (28 studies) [39,41–43,45–49,51,52,54,56–60,62–70,75]. *Auditory-linguistic* skills included skills relating to components of auditory or linguistic processing (e.g., listening to high frequency sounds, ability to listen to multiple speakers, utilising of semantic knowledge); whereas *compensatory skills* were suggested to be skills/actions deployed by an adults with HL to offset the suboptimal auditory signal arising from HL and/or the current acoustic conditions (e.g., planning, use of visual clues such as written text). Listening behaviours and skills were described as either having an *orientation towards self* (25 studies), such adjusting one's proximity to the speaker [39–45,49,51,52,56–59,61,62,64–67,69,70,75,78,80], or an *orientation towards others* (19 studies) [39,41,43,45,48–50,54,56,58–60,62,64–66,68,81]. For example, adults with HL described asking their communication partners to adapt their behaviour to facilitate interaction [47].

**Descriptive theme 2: External modifiers of perceived listening ability.** Data from 39 (84.8%) studies showed that perceived listening ability was often defined by adults with HL in terms of success/failure. This binary characterisation of listening lacked nuance, yet these data showed that adults with HL had a finely tuned awareness of the gradations of their ability. Participants were suggested to undertake differentiation or qualification of their perceived ability based on a range of external modifiers [44,48,57,58,64,78]. Participants described *environmental modifiers* of their perceived listening ability including the presence/absence of background noise, physical characteristics of the environment such as room size, reverberation, and lighting [46,57,64]. Ten studies described a participant's location relative to the sound source (i.e., distance to speaker) as influencing adults' with HL perceived listening ability. *Speaker characteristics* were also used by adults with HL to differentiate their listening abilities. Listening to a single speaker was perceived to be easier than listening to multiple speakers [47–49,60,70,77,78]. Female speakers, children and specific vocal characteristics of the speaker (e.g., soft speech, pitch, rate of speech, familiarity) were described by adults with HL as determiners of their perceived ability to listen [43,60,78]. Data from 16 studies suggested that *technical supports* such as hearing devices (i.e., hearing aids, FM systems and cochlear implants), smart phone technology, and captioning were also considered by adults with HL to be modifying influences [41,42,44,46,49–51,56–58,64,69,70,73,75,80]. Lastly, *information complexity* was identified by participants to be a determiner of their ability to listen [57,64]. Altogether,

appraisal of these external modifiers by adults with HL informed their decisions about the recruitment of the skills and the degree of flexibility needed to navigate communicative interactions [45,58].

**Descriptive theme 3: Psychosocial impacts of listening with a hearing loss.** Data from 37 (80.4%) studies described psychosocial outcomes of living with HL [39,41–47,49–51,53,54,56–59,61,63–70,72–75,77–80,82–84].Overall, HL was suggested to have negative impacts on *social activity*, participants' *emotional and physical well-being*, and *quality of life*. *Difficulty participating in social activities* was a recurring theme and was discussed in 21 studies [41,42,44,45,50,51,54,56–59,61,64–66,69,70,72,73,75,79].Participants reported frequently *withdrawing* from social activities or declining to participate due to difficulties with hearing and listening (nine studies),[41–43,49,57,58,65,66,72] and 20 studies reported participants' experiences of *social isolation* [41–46,49,50,53,54,56–59,61,66,69,70,82,84]. The studies also described a range of *emotional states* including, for example, *denial* (2 studies),[65,82] *disempowerment* (4 studies),[43,44,54,59] *embarrassment* (10 studies) [41–43,46,49,50,61,63,72,80], *grief* (2 studies), [43,46] and *stress* (8 studies) [42,43,51,59,64,72,80,83], as well as *rejection/exclusion* from social groups (6 studies) [41,43,44,46,54,56]. Participant experiences of *stigma* relating to HL were reported in 12 studies [39,42–45,54,57,58,61,65,82,83]. Eight studies suggested that *acceptance of hearing difficulties* and an ability to *adapt and accommodate* HL both within one's self-identity and, more practically, within their communicative interactions, was a determiner of an individual's psychosocial state [39,49,53,54,61,65,68,70]. Individuals who expressed a willingness to accept and accommodate their HL were suggested to have a more positive outlook, higher self-efficacy, increased participation and independence, and were agile with skill deployment to facilitate listening success within their communicative interactions.

**Descriptive theme 4: Communication partner perspectives, roles, and implications.** Data from 31 (67.4%) studies described either 1) the role/implications of communication partners from the perspective of adults with HL or 2) reported on communication partners' beliefs, attitudes, and perceptions of the listening abilities of the person with HL [39,41–47,49–54,56,58,59,62–64,66,68–70,72,75,77–80,82]. Codes reflected *specific skills*, mostly compensatory, deployed by communication partners and the *wider impacts of HL* on the relationship between adults with HL and their communication partners. Communication partners in the included studies were significant others (i.e., spouses, friends or adult children) or healthcare practitioners (HCPs). Coding within this theme suggested that communication was impacted by HL for both the adults with HL and their communication partners. Communication partners deployed a range of strategies to support the adults with HL including the use of *clear speech* (4 studies) [41,47,49,60], getting adults' with HL *attention* before speaking (2 studies) [41,49], and *relaying précised information* (7 studies) [43,45,47,56,58,68,82]. Eleven studies suggested that adults with HL took responsibility for ensuring their communication needs were known and requested that *communication partners adapt their behaviour* accordingly [39,41,43,45,49,54,56,59,60,64,66].

Four studies suggested that communication partners acted as *advocates* for their loved one with HL [43,45,52,53]. Communication partners were found to engage in high level surveillance by *monitoring and/or controlling conversations* (2 studies) [70,82] or acting as *mediators* between the adults with HL and their interlocutors (7 studies) [43,45,47,56,58,68,82]. Communication partners were suggested to intervene in conversations to *protect* their significant other with HL and facilitate communicative success (6 studies) [41,43,45,53,68,82]. This additional role was considered by participants, both adults with HL and communication partners, in eight studies to impact *relationships* (8 studies) [42–44,46,50,70,70,82], contributing to increased levels of *stress* (8 studies) [42,43,51,59,64,72,80,83], *loss of adults' with HL independence* (3 studies) [43,44,72], and relationship *breakdown* (2 studies) [43,53].

**Descriptive theme 5: Self-efficacy for listening.** Data from 15 (32.6%) studies described adults' with HL *self-efficacy* for listening [43–45,51,54,56–59,64,66,69,73,74,77]. Eight studies suggested that, overall, many (but not all) adults with HL perceive themselves *capable of listening*, despite experiences of communication breakdown [44,45,57,58,64,69,73,77]. Furthermore, study findings suggested participants did not consistently associate HL with the concept of *disability* or with *poor health* [22,44,54,56,77]. Rather, difficulty performing a particular listening-related skill was often perceived by adults with HL to be an inaccurate reflection of their innate capacity for listening. Instead, a performance decrement was attributed to external factors (e.g., directly to the HL or environmental factors) [44,45,58,73]. Six studies proposed that high self-efficacy was related to adults' with HL ability to implement listening skills that were *adaptive* and *proactive* [44,56,57,64,69,77]. When moments of *mismatch* between perceived capability and observed ability occurred, these were suggested to have negative psychosocial consequences for participants with HL [51,54,56,57,59,66,74].

**Descriptive theme 6: Cognitive load.** Data from 18 (39.1%) studies described listening for communication by adults' with HL as requiring additional investment of cognitive energy not required by individuals who do not have HL [40,43–46,48,50,56–58,60,64,66,71,72,78,82,85]. This energy requirement was described as *listening effort* in the included studies and was considered to result in listening-related *fatigue*. Three studies [41,57,58] described cognitive load as an increased requirement for *attention and concentration* when listening which participants suggested was something that happened *without conscious awareness* but was influenced by an individual's *motivation* to listen [57,58]. Five studies described attention and concentration in terms of participants' ability to successfully *divide their attention* between listening and a concurrent task [45,57,58,64,82]. Four studies suggested that the increased cognitive load required for listening was experienced and understood by participants in the context of the *dynamics of communication* [40,43,57,78]. In these studies, participants suggested that the increased need for *cognitive processing* when listening was experienced by participants as a *time lag* and difficulty keeping pace and *getting a turn* within in a conversational exchange. Ten studies suggested the increased processing demands experienced by adults with HL led to downstream *mental and physical fatigue* and *cognitive resource depletion* [40,46,48,50,56–58,60,64,72].

**Analytic themes: Liminality and reciprocity.** The six descriptive themes were suggested to be related by the two analytic themes of *liminality* and *reciprocity*. Fig 2 presents a hypothetical model of relationships between the descriptive themes based on the review findings. Firstly, we propose that adults with HL have insight into a range of listening-related skills providing these are supraliminal (i.e., above the threshold of conscious perception). The data

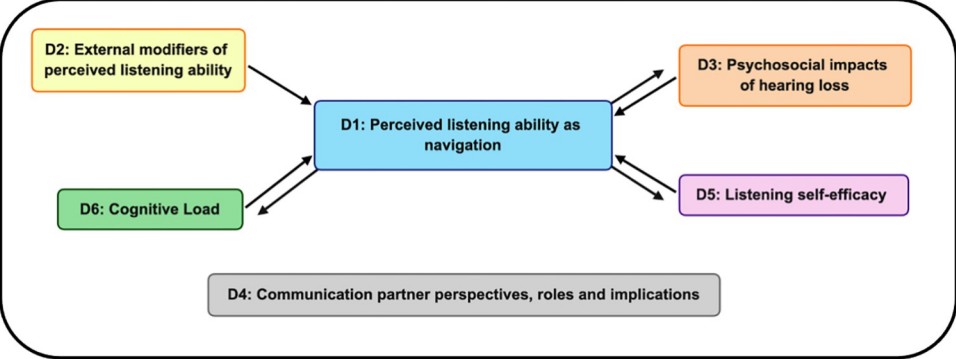

**Fig 2. Hypothetical model of relationships among the descriptive themes.**

suggested that adults with HL rely on external factors, as perceivable, supraliminal features, to grade their listening abilities against task difficulty. Within the context of listening for communication, adaptive or compensatory skills (e.g., communication techniques and behaviour modifications) that enable adults with HL to experience listening success were suggested to be a component of perceived listening ability. Secondly, we propose reciprocal relationships between the descriptive themes. For instance, adults' with HL perception of their listening abilities was informed by listening self-efficacy (i.e., perceived capability for listening). In turn, participants' performance (listening ability) in everyday communication situations was suggested to reinforce listening self-efficacy. Data from the contributing studies suggested that low listening self-efficacy had negative impacts on future listening-related performance, including skill selection and utilisation (i.e., selection of maladaptive v. adaptive listening-related behaviours). Social and psychological constructs were suggested to influence self-reported listening abilities and self-efficacy. In a reciprocal fashion, both were suggested to influence adults' with HL psychosocial status. Lastly, data suggested that participants considered listening effort and downstream fatigue effects to be influencers of their listening abilities and performance in real-world communication. In turn, skill selection and deployment were suggested by participants to contribute to the extent of their listening-related fatigue.

## Confidence in the review findings

An overall assessment of confidence in each descriptive theme was conducted by applying the GRADE CERQual approach. Confidence ratings ranged from "low" to "moderate". Themes D1-4,6 were appraised as "moderate confidence" and Theme D5 (Self-efficacy for listening) was appraised as "low confidence". A rating of "moderate confidence" suggested that a review finding was likely a reasonable representation of the phenomenon of interest and a rating of "low confidence" meant the review finding is likely an incomplete representation. Individual domain ratings and the confidence rating for each theme are summarised in Table 4. S2 Table presents results of the full GRADE CERQual evaluation.

## Discussion

This systematic review is the first qualitative meta-synthesis to examine perceived listening in HL from a metacognitive perspective. Through a review of primary qualitative studies, we explored adults' with HL and their communication partners conceptualisation of their listening abilities. Six descriptive themes were identified from the findings of 46 included studies. These themes described: listening abilities and component skills from the lived experience perspective, listening self-efficacy, the perceived requirement for additional cognitive resource allocation when listening, and the psychosocial impacts of listening with HL. A sixth theme capturing the implication of HL on communication partners provided an extended perspective on adults' with HL listening abilities in the context of interactive communication. Descriptive themes were linked through the higher-order analytic themes of liminality and reciprocity. CERQual grading of the descriptive themes were "low" (Theme D5—Self-efficacy for listening) and "moderate" confidence (Themes D1-4, D6), suggesting most themes were likely to be a reasonable representation of the available evidence.

Metacognition may be defined as "*knowledge* [emphasis added] of one's knowledge, processes, and cognitive and affective states; and the ability to *consciously and deliberately monitor and regulate* [emphasis added] one's knowledge, processes and cognitive and affect states." [14] (p.4) and reflects the analytic theme of liminality identified in this review. Themes D1, D2, D4, D5 and D6 suggest that adults' with HL and have knowledge of component skills and listening behaviours that they and their communication partners are required to adopt for

**Table 4. Assessment of confidence of review findings: Summary score per GRADE CERQual domain.**

| Theme | Theme label | Studies contributing to theme | Assessment of methodological limitations | Assessment of relevance[ii] | Assessment of coherence[iii] | Assessment of adequacy[iv] | Overall CERQual rating[v] |
|---|---|---|---|---|---|---|---|
| D1 | Perceived listening ability as navigation (within the dynamic, transactional space of communicative acts) | [39–52,54,56–81,84,85] | Moderate concerns | Moderate Concerns | No or very minor concerns | Minor concerns | Moderate confidence |
| D2 | External modifiers of perceived listening ability | [39–43,45–51,53,56–64,66,67,69–73,75–82,84] | Moderate concerns | Moderate concerns | No or very minor concerns | Minor concerns | Moderate confidence |
| D3 | Psychosocial impacts of hearing loss | [39,41–47,49–51,53,54,56–59,61,63–70,72–75,77–80,82–84] | Moderate concerns | Moderate concerns | No or very minor concerns | No or very minor concerns | Moderate confidence |
| D4 | Communication partner perspectives, roles and implications | [39,41–47,49–56,59,62–64,66,68–70,72,75,77–80,82] | Moderate concerns | Moderate concerns | No or very minor concerns | Moderate concerns | Moderate confidence |
| D5 | Self-efficacy for listening | [38,39,42,49,52,55,56,62,66–68,77,79,81,82] | Moderate concerns | Moderate concerns | Moderate-to-serious concerns | Moderate concerns | Low confidence |
| D6 | Cognitive Load | [40,43–46,48,50,56–58,60,64,66,71,72,78,82,85] | Moderate concerns | Moderate concerns | No or very minor concerns | Minor concerns | Moderate confidence |

CASP scoring of individual studies contributing to a theme.

[ii]Fit between original study aims, context and the review aims and context.

[iii]Fit between original study findings and review themes.

[iv]The extent to which themes are supported by quantitative evidence (number of studies and nodes supporting theme) and qualitative.

evidence (richness of data supporting a theme).

[v] Overall rating based on judgements for the four domains and scored as: High, moderate, low, or very low [34].

effective communication. Data from the contributing studies also suggested that adults with HL and their communication partners engage in monitoring and regulation through the deliberate adaptation of listening behaviours considered to increase the likelihood of communicative success. Similar findings have been reported in the second language learning literature. A qualitative study explored deployment of listening strategies (i.e., interest management, asking pre-questions, and elaboration strategies) by L2 learners and found that participants consistently reported more comprehensive understanding, deeper level of processing, and improved processing characteristics (e.g., sustaining attention and selective focus, better retention) when metacognitive strategies were appropriately aligned to the listening situation [86]. However, metacognition is known to be sensitive to listener characteristics. In a review exploring the contribution of cognition to speech-in-noise listening, listening strategy selection was found to depend upon listener characteristics such as age, hearing status, linguistic competency, educational attainment, and cognitive status [87]. Metacognitive judgements (i.e., monitoring which the authors define as one's awareness of cognitive processing) have been found to be largely preserved in older adults relative to younger adults but only to the extent that monitoring does not draw upon executive attention [88]. Further work is needed, building upon this body of work and the findings of this meta-synthesis, to understand the differential contribution of various listener characteristics on metacognitive knowledge, monitoring, and regulation in the context of HL.

Theme 3 (psychosocial impacts) suggested that adults with HL have a rich, nuanced knowledge of their emotional states in the context of HL. This finding is unsurprising given the established literature on the social and psychological consequences of HL. For example, hearing is a recognised risk factor for loneliness and social isolation, lower generic quality of life, and is associated with increased odds of psychological distress and utilisation of health services [89–91]. The significant psychosocial impacts reported by adults with HL across the included studies may be considered a reflection of their metacognitive beliefs. Metacognition may also account for the finding that self-efficacy (Theme 5) appeared to influence strategy deployment and coping. Self-Regulatory Executive Function (S-REF) theory proposes that metacognitive factors (e.g., beliefs about cognition, emotion and behaviour) can provoke and maintain states of psychological distress and predispose individuals towards maladaptive coping strategies (e.g., heightened self-focussed attention, threat monitoring, avoidance) which further compromise their psychological wellbeing [92]. Negative metacognitive beliefs have been associated with perceived stress and negative emotion in healthy volunteers and have been related to anxiety, depression and quality of life in patients with chronic conditions and their caregivers [93,94]. Although not explicitly stated, Barker et al., in a systematic review and qualitative meta-synthesis exploring the psychosocial experiences of people with HL and the communication partners, allude to the influence of metacognition, proposing that perceptions of self may influence the effect of HL and subsequent coping [95].

The influence of social factors on metacognition in individuals is well established in the literature. Frith (2012) reasons that explicit metacognition, which enables individuals to reflect on and justify their behaviour, is determined largely through social interactions with others and is a means by which individuals overcome the lack of direct access to underlying cognitive processes, including those recruited when listening [96]. Theme 4, which presented communication as a dynamic, reciprocal process with communication partners playing a key role in ensuring communicative success for the person with HL, underscores Frith's proposition that explicit metacognition is, in part, socially constructed. Recent work from Harris et al. (2019) applies this understanding to investigate memory collaboration in elderly couples. Findings from this study demonstrate that couples used established, sensitive and dynamic communication strategies to enhance recall [97]. A review by Cooke et al. also describes overt modifications to speech directed at listeners with HL aimed at promoting audibility, increasing coherence, enhancing linguistic information or decreasing cognitive effort associated with the listening task [98]. The present trend to include interlocutors or observers to enhance the ecological validity of hearing research suggests there is a growing awareness by cognitive hearing scientists of communication partners' influence on adults with HL self-appraisal and consequently their self-management of real-world listening (for examples, see [99–101]). Understanding the communication mechanisms in interpersonal relationships will be key if measurement of communication ability is to be both valid and reliable.

## Strengths and limitations

This study has some limitations. The GRADE CERQual approach, despite providing a systematic process for appraising quality of the findings, did not provide sufficiently operationalised acceptability criteria to support reviewer judgments, with an associated risk of bias. To address this limitation, a second member of the review team (KN) reviewed all GRADE CERQual ratings with the involvement of a third reviewer (IB) to reach consensus when discrepancies were identified. Ideally, two reviewers would independently undertake a GRADE CERQual assessment of evidence quality; however, this was not possible in the current study due to resource limitations. High kappa and ICC values for the CASP assessment of articles' methodological

quality provides supporting evidence of strong inter-rater agreement and reliability between the two reviewers.

Second, limiting searches to studies published in English-language journals raises the possibility of publication bias. Only one article published in a language other than English was identified through the electronic database searches which suggests relevant articles were likely to have been published in English and therefore identified by the searches. Third, this review presents a secondary analysis of primary data; therefore, any methodological limitations of the included studies were carried forward to this review. For example, the relationship between researcher and participants was not reported adequately for most of the included studies. Inadequate accounts of reflexivity raised uncertainty about the handling of researcher bias and downgrading of a study's methodological quality. GRADE CERQual assessment of the review findings enabled these methodological limitations of the primary studies to be considered in the context of the quality of the overall findings of this review. Last, it was not possible to undertake sub-analyses to explore age-related difference as data in the primary studies did not explicitly differentiate between younger v older listeners. In addition, few articles included participants with mild or moderate hearing loss (MMHL); therefore, it was not possible to mine the experiences of listening for this group. Lack of representation across the full range of hearing loss has implications for conceptual operationalisation of a PRO and for ensuring complete construct coverage, particularly at the extreme of the measurement continuum (i.e., perceived listening ability at mild or profound levels of hearing loss).

## Implications for research and practice and directions for future research

This review, as a secondary analysis of primary data, provided a synthesis of current research exploring metacognition of listening from a lived experience perspective of hearing loss. Studies are now required to validate the proposed themes in primary qualitative studies. The review findings contribute an alternative perspective to the body of literature on perceived listening ability and metacognition which has drawn extensively from studies of second language learners in an educational context. Further work to explore perceived listening ability across subgroups of the population adults with HL (e.g., single-sided deafness, older v younger adults) is now needed to ensure conceptual representativeness and construct coverage. Further work is also needed to investigate mediating role of listener characteristics on metacognitive knowledge, processes and behaviour when a hearing loss is present. Findings from this review will be used to inform the content and design of a new patient reported outcome measure (PROM) of perceived listening ability in hearing loss. Findings may also be used to inform trial design by ensuring patient-reported outcome (PRO) selection that is representative of the knowledge, processes and experiences of listening considered important to adults with HL.

Conceptualisation of perceived listening ability in a metacognitive framework could inform the development of supportive interventions designed to mitigate the wider impacts of HL. The findings could also inform the design of trials to investigate the efficacy and effectiveness of for aural rehabilitation (AR) interventions. As a heuristic framework, metacognition is relevant because the underlying mechanisms of AR are not yet well developed and high-quality evidence of AR benefit is limited [102]. Support for the implementation of a metacognitive framework in AR may be drawn from the wider metacognition literature. Goal-based metacognitive strategy interventions have been shown to improve attention, executive functioning, self-awareness, and social communication in adults with traumatic brain injury (TBI) [103–105]. Second language (L2) learners receiving explicit communication strategy instruction have been found to have greater metacognitive awareness, defined as conscious awareness of communication strategies, and increased frequency of use of "high-quality" communication

strategies compared with controls [106]. In the AR setting, a metacognitive framework that conceptualises the components of functional listening could guide the identification and targeting of specific skills, behaviours, knowledge and processes for inclusion in a self-management intervention. Awareness of the contribution of cognitive and affective states to perceived listening ability could support counselling and information giving in the context of clinical audiology practices. A metacognitive understanding of relationships between hearing loss and its social, psychological and emotion impacts could provide opportunities for intervention through metacognitive modification and training.

## Conclusions

This meta-synthesis of qualitative studies explored perceived listening ability and hearing loss from a metacognitive perspective. By considering individuals' knowledge and understanding of their own listening abilities and their capacity to monitor and regulate their listening behaviour, the findings underscore the dynamic communication complex in which real-world listening is situated. As a heuristic, the findings provide a more holistic framing of listening function with potential to inform the design of empirical studies with greater ecological validity, concept elicitation for new patient-reported outcome measures, and the development of supportive interventions, that enable individuals living with hearing loss to more effectively manage the dynamic and unpredictable communication demands of daily life.

## Supporting information

**S1 Checklist. PRISMA Checklist.** Preferred Reporting Items for Systematic reviews and Meta-Analyses extension for Scoping Reviews (PRISMA-ScR) Checklist.
(DOCX)

**S1 Table. Characteristics of the included studies (N = 46).**
(DOCX)

**S2 Table. GRADE CERQual evaluation.**
(XLSX)

**S1 File. Medline search strategy.**
(DOCX)

**S1 Dataset. Extracted data and quality assessment.**
(XLSX)

## Acknowledgments

The authors thank JvB, Information Specialist at Macquarie University, for their help developing the search strategy and BH for their help with technical editing and manuscript preparation.

## Author Contributions

**Conceptualization:** Sarah E. Hughes, Anne Steyn, Katie Neal.

**Data curation:** Sarah E. Hughes, Katie Neal.

**Formal analysis:** Sarah E. Hughes, Isabelle Boisvert, Catherine M. McMahon, Katie Neal.

**Funding acquisition:** Sarah E. Hughes.

**Methodology:** Sarah E. Hughes, Katie Neal.

**Project administration:** Sarah E. Hughes, Katie Neal.

**Resources:** Sarah E. Hughes.

**Writing – original draft:** Sarah E. Hughes.

**Writing – review & editing:** Sarah E. Hughes, Isabelle Boisvert, Catherine M. McMahon, Anne Steyn, Katie Neal.

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
