## [Decision Letter · Decision Letter 0]

17 Aug 2022

PONE-D-22-16725Perceived listening ability and hearing loss: systematic review and qualitative meta-synthesisPLOS ONE

Dear Dr. Hughes,

Thank you for submitting your manuscript to PLOS ONE. After careful consideration, we feel that it has merit but does not fully meet PLOS ONE’s publication criteria as it currently stands. Therefore, we invite you to submit a revised version of the manuscript that addresses the points raised during the review process.

We look forward to receiving your revised manuscript.

Kind regards,

Rohit Ravi, Ph.D.

Academic Editor

PLOS ONE

Journal Requirements:

“SEH receives funding from the National Institute for Health Research (NIHR) Applied Research Collaboration (ARC) West Midlands and UK Research and Innovation (UKRI), UK SPINE, and declares personal fees from Aparito Limited outside the submitted work. IB declares funding from Birmingham Health Partners Centre for Regulatory Science & Innovation / UK SPINE outside the submitted work. All other authors have no competing interests to declare.”

Additional Editor Comments:

Dear Authors I have received the comments from reviewers there are minor changes suggested, do look into the same and submit.

Reviewers' comments:

Reviewer's Responses to Questions

**Comments to the Author**

1. Is the manuscript technically sound, and do the data support the conclusions?

Reviewer #1: Yes

Reviewer #2: Yes

2. Has the statistical analysis been performed appropriately and rigorously? 

Reviewer #1: Yes

Reviewer #2: Yes

3. Have the authors made all data underlying the findings in their manuscript fully available?

Reviewer #1: Yes

Reviewer #2: Yes

4. Is the manuscript presented in an intelligible fashion and written in standard English?

Reviewer #1: Yes

Reviewer #2: Yes

5. Review Comments to the Author

Reviewer #1: Good work. More comments in the attached file. May be good to review on the clarity sought. The themes analyzed, but coding in the table can give clarity. Also, good to explain how the researcher and participants influence not clear in many studies how it will influence the outcomes of the study.

Reviewer #2: “Perceived listening ability and hearing loss: a systematic review and qualitative metasynthesis” was an interesting topic to study. The highlighting part of the study was the number of studies over the map. The authors appeared to be well-versed with the information about the idea behind the study. Authors have put efforts to represent the study which makes it easy to follow. I appreciate authors have included implications and limitations however withdrawing the theme in these two sections could be considered.

6. PLOS authors have the option to publish the peer review history of their article (what does this mean?). If published, this will include your full peer review and any attached files.

Reviewer #1: No

Reviewer #2: No

---

## [Author Response · Author response to Decision Letter 0]

26 Sep 2022

Response to editorial requests and reviewers’ comments:

Comments from the editorial team

Thank you, we have consulted the guidance and checked the manuscript carefully to ensure it complies with PLOS ONE’s style requirements. Any changes are noted in the tracked changes and clean versions of the manuscript. 

a)        Please clarify the sources of funding (financial or material support) for your study. List the grants or organizations that supported your study, including funding received from your institution. 

b)        State what role the funders took in the study. If the funders had no role in your study, please state: “The funders had no role in study design, data collection and analysis, decision to publish, or preparation of the manuscript.” 

c)        If any authors received a salary from any of your funders, please state which authors and which funders. 

d)        If you did not receive any funding for this study, please state: “The authors received no specific funding for this work.” 

Thank you for your queries which we have responded to below: 

a) This work was supported by Cochlear Limited (Funder Ref: Functional Listening for Communication Project). 

c) SEH and KN received funding from Cochlear Limited to undertake this work. 

d) Not applicable- see c) above

“SEH receives funding from the National Institute for Health Research (NIHR) Applied Research Collaboration (ARC) West Midlands and UK Research and Innovation (UKRI), UK SPINE, and declares personal fees from Aparito Limited outside the submitted work. IB declares funding from Birmingham Health Partners Centre for Regulatory Science & Innovation / UK SPINE outside the submitted work. All other authors have no competing interests to declare.”

We have updated the competing interests statement as requested and have included the statement in the cover letter above. 

In your Data Availability statement, you have not specified where the minimal data set underlying the results described in your manuscript can be found. PLOS defines a study's minimal data set as the underlying data used to reach the conclusions drawn in the manuscript and any additional data required to replicate the reported study findings in their entirety. All PLOS journals require that the minimal data set be made fully available. For more information about our data policy, please see http://journals.plos.org/plosone/s/data-availability.

Important: If there are ethical or legal restrictions to sharing your data publicly, please explain these restrictions in detail. Please see our guidelines for more information on what we consider unacceptable restrictions to publicly sharing data: http://journals.plos.org/plosone/s/data-availability#loc-unacceptable-data-access-restrictions. Note that it is not acceptable for the authors to be the sole named individuals responsible for ensuring data access. We will update your Data Availability statement to reflect the information you provide in your cover letter.

Thank you for your comment. We wish to refer the editors to our previous data availability statement which is consistent with data availability statements include with systematic reviews published by PLOS ONE. 

The statement is as follows: “All relevant data may be found within the manuscript and its Supporting Information Files. All data sources are publicly available, peer-reviewed journal articles listed in the article’s references.”

Thank you. We have updated the manuscript accordingly. Lines 947-953, pp45-46

We have checked the reference list and confirm that no changes were made. 

Comments from Academic Reviewers

Reviewer #1: Good work. More comments in the attached file. May be good to review on the clarity sought. The themes analyzed, but coding in the table can give clarity. Also, good to explain how the researcher and participants influence not clear in many studies how it will influence the outcomes of the study.

Thank you for your comments. We have presented exemplars of the coding and themes in Table 3 for illustration purposes. We have also discussed reflexivity and researcher bias in the discussion by adding the following:

“For example, the relationship between researcher and participants was not reported adequately for most of the included studies. Inadequate accounts of reflexivity raise uncertainty about the handling of researcher bias and lead to the downgrading of a study’s methodological quality.” Lines 562-564, p 31.

Reviewer #2: “Perceived listening ability and hearing loss: a systematic review and qualitative metasynthesis” was an interesting topic to study. The highlighting part of the study was the number of studies over the map. The authors appeared to be well-versed with the information about the idea behind the study. Authors have put efforts to represent the study which makes it easy to follow. I appreciate authors have included implications and limitations however withdrawing the theme in these two sections could be considered.

Thank you for your comment and your appreciation of our efforts. Referring to “withdrawing the theme”, we were unsure of the reviewer’s meaning; however, we have interpreted this sentence to mean greater emphasis on “perceived listening ability” within the limitations and implications sections. We have revised these sections and would be happy to refine further if necessary and following clarification of the reviewer’s comment. 

“This review, as a secondary analysis of primary data, provided a synthesis of current research exploring metacognition of listening from a lived experience perspective of hearing loss. Studies are now required to validate the proposed themes in primary qualitative studies. The review findings contribute an alternative perspective to the body of literature on perceived listening ability and metacognition which has drawn extensively from studies of second language learners in an educational context. Further work to explore perceived listening ability across subgroups of the population adults with HL (e.g., single-sided deafness, older v younger adults) is now needed to ensure conceptual representativeness and construct coverage. Further work is also needed to investigate mediating role of listener characteristics on metacognitive knowledge, processes and behaviour when a hearing loss is present. Findings from this review will be used to inform the content and design of a new patient reported outcome measure (PROM) of perceived listening ability in hearing loss. Findings may also be used to inform trial design by ensuring patient-reported outcome (PRO) selection that is representative of the knowledge, processes and experiences of listening considered important to adults with HL.” Lines 577-591, pp31-32

The researchers have selected a good topic and the fact of the number of articles they have included justifies the need for a systematic review with metasynthesis. Have done good effort on including relevant studies and search engines. The method adapted justifies the research question and well explained. Author may give explanation on the need to include Google scholar and specifically only first 200 records). If the editor justifies the need to be included that comment in the manuscript that is fine, if not, I am ok to seek clarification.

When selecting search engines, we referred to the work of Bramer et al. (2017) who undertook an exploratory study to identify optimal database combinations for literature searches in systematic reviews. Based on the study findings, the authors recommended that to optimise retrieval, systematic reviews should search at least Embase, MEDLINE, Web of Science, and Google Scholar as a minimum requirement to guarantee adequate and efficient coverage. Based on this recommendation and, working in collaboration with an Information Specialist, we opted to include Google Scholar in the selection of electronic databases for this review. We have included the following in the manuscript: “Database selection, including Google Scholar, was guided by recommendations for optimal yields proposed by Bramer et al.[23] Lines 156-157, p6.

In the results section, the authors have made a good attempt to summarize and present in well structured sentences. However, in table 3 the author has given quality appraisal of the articles. It is observed that, for the 6th question, on relationship between researcher and participants many studies the relation was not clear. In a research this relation is important. May be good idea if the author, although mentioned, good to comment how this can influence on the interpretation of the results.

Thank you for your comment and we agree that this is an important point with relevance to the interpretation of findings. The methodological limitations (including reflexivity and researcher bias) of the included studies are appraised as part of the GRADE CERQual assessment to derive a rating of confidence of the review findings. The GRADE CERQual assessment is published in full in Supporting Information File S2 Table. We have also included in the discussion the following: 

“For example, the relationship between researcher and participants was not reported adequately for most of the included studies. Inadequate accounts of reflexivity raise uncertainty about the handling of researcher bias and lead to the downgrading of a study’s methodological quality.” Lines 562-564, p. 31

In the themes table 4, may be good to include the codes instead of bullets to get more clarity.

Thank you for your comment. To clarify, the bullet points are first-level codes as stated in the table header “Examples of initial codes” We have removed the bullet points and changed the head to “Initial code” to improve clarity. Table 3, p 16.

It is also appreciated the author has presented the discussion well and highlighted the limitations of the study. It will be good the implication/conclusions/themes drawn in the study are taken further.

Thank you to the reviewer for suggesting elaboration of the conclusion and potential impact of the study. In the manuscript we have provided readers with several potential applications/implications of the findings. We have considered the reviewers requests and have made the following revisions:

“This review, as a secondary analysis of primary data, provided a synthesis of current research exploring metacognition of listening from a lived experience perspective of hearing loss. Studies are now required to validate the proposed themes in primary qualitative studies. The review findings contribute an alternative perspective to the body of literature on perceived listening ability and metacognition which has drawn extensively from studies of second language learners in an educational context. Further work to explore perceived listening ability across subgroups of the population adults with HL (e.g., single-sided deafness, older v younger adults) is now needed to ensure conceptual representativeness and construct coverage. Further work is also needed to investigate mediating role of listener characteristics on metacognitive knowledge, processes and behaviour when a hearing loss is present. Findings from this review will be used to inform the content and design of a new patient reported outcome measure (PROM) of perceived listening ability in hearing loss. Findings may also be used to inform trial design by ensuring patient-reported outcome (PRO) selection that is representative of the knowledge, processes and experiences of listening considered important to adults with HL.” Lines 577-591, pp31-32

---

## [Editor Report · Decision Letter 1]

4 Oct 2022

Perceived listening ability and hearing loss: systematic review and qualitative meta-synthesis

PONE-D-22-16725R1

Dear Dr. Hughes,

We’re pleased to inform you that your manuscript has been judged scientifically suitable for publication and will be formally accepted for publication once it meets all outstanding technical requirements.

Kind regards,

Rohit Ravi, Ph.D.

Academic Editor

PLOS ONE

Additional Editor Comments (optional):

Dear Authors, the changes are satisfactory.
---

## [Editor Report · Acceptance letter]

17 Oct 2022

PONE-D-22-16725R1 

Perceived listening ability and hearing loss: systematic review and qualitative meta-synthesis 

Dear Dr. Hughes:

I'm pleased to inform you that your manuscript has been deemed suitable for publication in PLOS ONE. Congratulations! Your manuscript is now with our production department. 

Kind regards, 

on behalf of

Dr. Rohit Ravi 

Academic Editor

PLOS ONE